# Socioeconomic and Demographic Factors for Spousal Resemblance in Obesity Status in China

**DOI:** 10.3390/healthcare8040415

**Published:** 2020-10-21

**Authors:** Xuejiao Chen, Xueqi Hu, Songhe Shi, Qingfeng Tian

**Affiliations:** 1Department of Social Medicine and Health Management, College of Public Health, Zhengzhou University, Zhengzhou 450001, China; chenxuejiao0516@outlook.com; 2Department of Epidemiology and Health Statistics, College of Public Health, Zhengzhou University, Zhengzhou 450001, China; 18513506975@163.com (X.H.); zzussh@126.com (S.S.)

**Keywords:** body mass index, overweight, obesity, socioeconomic factors, family health, China

## Abstract

Introduction: The purposes of this study were to explore the resemblance in the weight status within couples with different family contextual factors and analyze the influence of the level of overweight or obesity of a spouse on that of the other spouse. Methods: The data were from the sixth National Health Service Survey of Henan Province in 2018. After screening, 7432 eligible couples were finally included. Socioeconomic and demographic factors were compared by the χ^2^ test or nonparametric test. The difference in the body mass index (BMI) of spouses was assessed by a t-test. The Pearson correlation coefficient and kappa value were used as indicators of consistency in weight status. A logistic regression analysis was used to further explore the effect of a spouse’s level of overweight/obesity on that of the other spouse. Results: The results show that the prevalence of overweight/obesity in couples aged 20 or older is 33.76%. The Pearson correlation coefficient of the BMI within couples was 0.102 (95% CI: 0.076–0.120). The kappa coefficients suggested a low resemblance in the weight status within couples (k = 0.049, 95% CI: 0.031–0.069). Besides, the influence of the overweight/obesity status of the wives on that of the husbands (odds ratio (OR) = 1.411, 95% CI: 1.309–1.521) was slightly higher than that of the husbands on that of the wives (OR = 1.404, 95% CI: 1.302–1.514). Conclusions: We found that there was a moderate but significant resemblance in the body weight status between spouses, especially elderly couples with a low education level in rural areas. Health education activities for couple interventions can have a good effect of intervention.

## 1. Introduction

Marital status is a social factor that is significantly related to individuals’ health and mortality [1]. Previous studies have shown that compared to single people, married people are more likely to be overweight or obese, eat more and have lower levels of physical activity [2,3,4,5,6]. The reason for this difference may lie in the positive or negative impacts of changes in the environments or lifestyles that occur after marriage on the couple’s health. Therefore, it is particularly important to understand the impacts of different socioeconomic conditions and demographic factors on the health of married couples.

Overweight and obesity are risk factors for numerous diseases, including ischemic heart disease, diabetes, and certain forms of cancer [7,8,9,10]. In particular, being overweight or obese is associated with a high mortality, disability, and poor quality of life [11,12,13].

Although there is clear evidence on the presence of spousal resemblance in weight status, little is known about how family contextual factors influence the degree of spousal resemblance in BMI. An individual’s BMI is dependent on many factors, such as age, education, food habits and sedentary lifestyles [14,15,16]. Spouses under the same family socioeconomic condition share resources and stresses in life, which in turn contribute to obesity-related behaviors related to eating and one’s health status. Identification of the family contextual factors associated with spousal resemblance may help identify families/couples needing assistance [17].

Using the sixth National Health Service Survey data collected in 2018 in Henan Province, we analyzed the absolute and relative indicators of BMI and overweight/obesity consistency of married couples to identify the characteristics of couples with a high consistency. In addition, we further explored the influence of different characteristics on the level of overweight/obesity of husbands or wives and their spouses to provide ideas for targeted measures to reduce the rate of overweight/obesity. 

## 2. Methods

### 2.1. National Health Service Survey

The data are from the sixth National Health Service Survey that was administered in Henan Province in 2018. The National Health Service Survey (NHSS) is an ongoing nationally representative survey of the civilian noninstitutionalized population that has been conducted every five years since 1993 using a multistage stratified cluster random sampling design. During the interviews for the NHSS, household respondents reported demographic characteristics, health conditions, habits and customs, health insurance coverage and employment for all household members. 

### 2.2. Study Population

With the survey, data from 10,860 households in Henan were collected in 2018. Individuals with diseases or health conditions that seriously affect body weight, such as pregnancy, HIV/AIDS, cancer, diabetes and thyroid disease, were excluded. After excluding individuals without a specific spouse identification code, couples who were married but separated and couples with inconsistent marital status reports, we initially obtained data from 7527 couples. We further excluded 95 spouses whose BMI values were extremely high or low (<15 and >42, 1.5 interquartile range (IQR) was lower than the bottom BMI quartile, and 3 IQR were higher than the top BMI quartile). The final study population included 14,864 married subjects, that is, 7432 adult, opposite-sex, cohabiting, and relatively healthy couples.

### 2.3. Key Study Variables

#### 2.3.1. Outcome Variables

Body mass index (BMI) was calculated as the self-reported body weight (kg) divided by the square of the subject’s height (m^2^). Underweight, normal weight, overweight and obesity were defined as BMI < 18.5 kg/m^2^, 18.5 kg/m^2^ ≤ BMI < 25.0 kg/m^2^, 25.0 kg/m^2^ ≤ BMI <30.0 kg/m^2^ and BMI ≥ 30.0 kg/m^2^, respectively. Overweight/obesity was defined as BMI ≥ 25 kg/m^2^ [18]. We combined the overweight and obesity groups because the prevalence of BMI ≥ 30 kg/m^2^ was very low (4.29% among the respondents in this study).

#### 2.3.2. Individual-Level Factors

These individual-level factors included demographics (age, marital status, rural/urban residence, childbearing situation), socioeconomic characteristics (years of education and employment status) and lifestyle factors (current smoking habit, current drinking habit and habitual physical activity). The marital statuses included unmarried, married, widowed and other unspecified situations. The childbearing situation included having children and not having children. The employment statuses included employed, retired, never worked, and unemployed. The habit parameters of couples in daily life, such as smoking status, drinking habits, and exercise status are all dichotomy issues, reported by the household respondent to the following questions: “Do you smoke now?”, “Have you drink alcohol or have had alcoholic drinks in the past 12 months?” and “Do you now spend half an hour or more in moderate or vigorous physical activity at least three times a week?”.

#### 2.3.3. Couple-Level Variables

The couple-level variables were developed by combining the spouses’ individual characteristics. The mean age of a couple was categorized as 20–29, 30–39, 40–49, 50–59, 60–69 or ≥70 years. The combination of the two spouses’ unemployment statuses was categorized as “both unemployed”, “only husband unemployed”, “only wife unemployed”, and “both employed.” The couples’ habitual physical activity statuses were combined and categorized as both, only husband, only wife, or neither of the spouses performing habitual physical activity. Other couple-level variables were classified similarly.

### 2.4. Statistical Analysis

All continuous variables are presented as the mean ± standard deviation. The categorical data are summarized as the count and percentage. The comparisons of the husbands’ and wives’ socioeconomic or demographic factors were performed by the χ^2^ test or nonparametric test.

A t-test was used to compare the difference in BMI between the husband and wife in each subgroup. Pearson’s correlation coefficients between the spouses’ BMIs were estimated by regressing the standard scores of the husbands’ BMIs on the standard scores of the wives’ BMIs. For the categorical outcomes, the kappa coefficients for weight status (normal weight, overweight, and obese) between spouses were calculated to measure the level of crude agreement. Finally, an individual-level logistic regression was used to examine how an individual’s characteristics and his or her spouse’s characteristics were associated with his or her level of overweight/obesity. An individual’s and his or her spouse’s individual-level lifestyles and socioeconomic status (SES) were included in the model together to study the differences in the associations between spouses.

Statistical analyses were performed using the statistical software package (IBM SPSS statistics version 23). All reported *p*-values were two-sided, with *p* < 0.05 being considered statistically significant.

## 3. Results

### 3.1. Description and Comparison of the Basic Information between the Husbands and Wives

The characteristics of the husbands and wives are shown in Table 1. There was a significant difference in the BMI groups and overweight/obesity rate between husbands and wives (*p* < 0.001), but there was no significant difference in the obesity rate (*p* = 0.311). Among married couples in China, the husbands are older and have a higher education level; the wives’ employment rate and smoking and drinking rates are lower than those of the husbands. The differences in residence, employment, having children and the prevalence of diabetes were statistically significant (*p* < 0.05), but there was no significant difference in habitual physical activity (HPA more than once a week) and hypertension (*p* > 0.05).

### 3.2. Absolute Similarity and Relative Association in Weight Status among Chinese Couples

Table 2 shows the absolute similarity of the BMI among Chinese married spouses, stratified by the couple-level characteristics. The average BMI of the respondents was 23.84 ± 3.38 kg/m^2^ (husbands: 23.96 ± 3.33 kg/m^2^; wives: 23.72 ± 3.44 kg/m^2^). There was a significant difference in the BMI of couples aged 20–49, but there was no significant difference in the BMI of couples aged 50 or older. For all individuals, the BMI of the 40- to 59-year-old group was higher than that of the 20- to 29-year-old group.

There was a significant difference in the BMI between urban and rural couples (*p* < 0.001), and the BMI of urban couples was higher than that of rural couples (*p* < 0.001). The wives had a significantly lower BMI than the husbands in urban and nonsmoking couples and couples involving spouses who both completed primary school or a higher level of education, were both employed or only the wife was employed, both drank alcohol or only the husband drank alcohol, and both performed HPA (*p* < 0.05). In couples where both spouses have no children or both have children, the wife’s BMI is significantly lower than the husband’s (*p* < 0.05).The wives had a significantly higher BMI than the husbands in couples where only the husband completed primary school or a higher level of education or only the wife drank alcohol (*p* < 0.05), and couples with no children have a lower BMI than couples with children. The BMI of the husband was higher than that of the wife in the couples without hypertension/diabetes or only the husband had hypertension/diabetes (*p* < 0.001). Wives with hypertension/diabetes had a higher BMI than husbands without hypertension/diabetes (*p* < 0.001).

Table 2 illustrates the relative association in weight status among the Chinese married spouses, stratified by the couple-level characteristics. The Pearson correlation coefficient between wives’ and husbands’ BMI was 0.102 (95% CI: 0.076–0.120). The correlation of BMI within couples aged 60 or older was higher than that within couples aged 40–59, while that of couples aged 20–39 was not significant. The correlation of BMI within rural couples was higher than that within urban couples. There was a relatively high correlation in the BMI within diabetic couples (r = 0.323, 95% CI: 0.077–0.481). The kappa coefficients suggested a low resemblance in the weight status (underweight, normal weight, overweight, and obese) between spouses (k = 0.049, 95% CI: 0.031–0.069). 

### 3.3. Association between the Level of Overweight/Obesity and an Individual’s/His or Her Spouse’s Characteristics

The differences in characteristics and probabilities of overweight or obesity between an individual and his or her spouse are shown in Table 3. After the model was adjusted for covariates, the influence of the level of overweight or obesity of the wife on that of the husband (OR = 1.411, 95% CI: 1.309–1.521) was slightly higher than that of the husband on that of the wife (OR = 1.404, 95% CI: 1.302–1.514). The possibility of overweight or obesity of the husband decreased by 3.7% every year with increasing age. 

Urban husbands were 24.3% more likely to be overweight or obese than the rural husbands. Wives with urban husbands were 28.0% more likely to be overweight or obese than those with rural husbands. Among all wives in this survey, the more educated wives were less likely to be overweight or obese. Among couples, the higher the education level of husbands, the more likely their wives were to be overweight or obese. The odds of being overweight or obese was increased in husbands whose wives had a higher education level, compared with husbands whose wives had the lowest education level.

Compared with the unemployed husbands, the working husbands were 14.8% less likely to be overweight or obese; compared with the unemployed wives, the working wives were 17.9% less likely to be overweight or obese. Wives whose spouses had jobs were 1.161 times more likely to be overweight or obese than those whose spouses were unemployed. Compared with husbands without children, husbands with children were 1.1% more likely to be overweight or obese; compared with wives without children, wives with children were 14.8% more likely to be overweight or obese. Husbands whose spouses have children were 1.113 times more likely to be overweight or obese than those whose spouses do not have children, women whose spouses have children were 1.231 times more likely to be overweight or obese than those whose spouses do not have children.

Compared with nonsmoking husbands, smoking husbands were 31.6% less likely to be overweight or obese, while drinking husbands were 43.2% more likely to be overweight or obese than nondrinking husbands. In couples involving a husband who drinks alcohol, the wives’ risk of being overweight or obese increases by 20.5%. Husbands who exercised more than once a week were 8.6% more likely to be overweight or obese than those who did not exercise more than once a week. 

Regardless of his or her sex, individuals with hypertension were more likely to be overweight or obese than those without hypertension. Husbands with diabetes were 19.8% more likely to be overweight or obese than those without diabetes. Wives whose spouses had diabetes were 1.390 times more likely to be overweight or obese than those whose spouses did not have diabetes.

## 4. Discussion

This population-based, cross-sectional study reveals spousal resemblance in the weight status of couples with consideration of their family’s background and socioeconomic status in Henan, China. According to our results, the spousal resemblance in the weight status in combination with the actual BMI of husbands and wives needs to be considered, as the two dimensions of resemblance, relative-scale indices and absolute-scale indices do not necessarily coincide with each other.

The results of this study show that the prevalence of overweight/obesity in married adults aged 20 or older in China is 33.76% (husbands: 35.55%; wives: 31.95%). More than half (54.49%) of couples had at least one partner who was overweight or obese. The prevalence of overweight and obesity in married adults in this study is higher than that in Japan (married women: 10.3%) [19] and lower than that in the United States (husbands: 74.4%; wives: 52.0%) [17], India (married women: 76.9%) [20] and European countries (married adults: 49.5%) [21].

Some studies have shown that spousal similarity decreases with marriage duration [22,23,24]. In contrast, we found that the spousal correlation in the BMI of elderly couples was higher than that of young couples. Previous research has shown that couples who cohabitate for longer periods of time are more likely to show concordance in behaviors related to obesity, such as low levels of physical activity and high levels of sedentary behavior [25].

Note that the marriage duration data were not included in this study, but the average age of the couples can be used as a rough indicator. With an increasing average age, the BMI of the couples first increased and then decreased. Thus, the amount and direction of the BMI changes may have an impact on the consistency of BMI within couples. The results of this study show that the BMI of husbands aged 50–59 is decreasing, while that of wives in the same age group is increasing. Therefore, the consistency of BMI between spouses in this age group is relatively low. However, the BMIs of the couples aged 60 or older decreased with an increasing average age, and the difference in the average BMI became more similar [17], so the consistency in the elderly couples’ BMIs increased.

This study found that the couples involving spouses who were both living in rural locations, were both unemployed, both had completed primary school or had a lower level of education and both did not perform HPA had a higher spousal resemblance. Compared with the full-time working, highly educated or urban couples, these couples may need to work together for farming or recreation, leading them to have more similar eating habits or lifestyles. However, these habits with stronger spousal resemblance are not necessarily good for one’s health, and generally, unhealthy behaviors and lifestyles are easier to share between husbands and wives [17]. In this survey, the BMI of couples with children was higher than that of couples without children, and the increase in overweight or obesity of wives in families with children was greater than that of husbands. This is related to the loss of sleep and physical exercise caused by the wife’s childcare [26].

Studies have shown that men who smoke have a lower risk of becoming overweight or obese than men who do not smoke, but drinking alcohol increases the risk of being overweight or obese for married men themselves and their spouses [27]. In general, married men who have drinking habits tend to become overweight or obese. Because of social interactions or pressure, most of these men have bad eating habits, such as excessive oil and salt intake, overeating or an irregular diet [28]. According to previous studies, people who are overweight or obese are considered to be at high risk of hypertension, hyperglycemia and hyperlipidemia, which are usually accompanied by obesity [29]. Similar conclusions have been drawn in this study: individuals with hypertension or diabetes are more likely to become overweight or obese than those without hypertension or diabetes. In addition, relevant studies and the literature have proposed and confirmed that poverty and food insecurity were strongly associated with obesity [30,31,32].

Some strengths of our study should be noted. First, with the national health service survey administered in Henan Province, this study adopted multistage stratified cluster random sampling to collect data from families. The questionnaire design and sampling method are scientific and rigorous, and the survey response rate and data quality were high. Second, we studied spouse similarity from absolute and relative perspectives, stratified each variable at the couple level, and analyzed the differences in the BMI, correlations and consistency values of each subgroup. Third, we also analyzed the differences in the risk of overweight or obesity of husbands and wives or their spouses caused by various socioeconomic factors.

Some limitations of this study should also be noted. Because of the cross-sectional nature of our study and the interdependency of many of the variables, no definite conclusions on the temporal or causal effects among the independent and dependent variables can be made. In addition, this survey was conducted on a household basis, and the family members (including household head, spouse, children, daughter-in-law or son-in-law, parents, grandparents, etc.) were not assessed individually. In Chinese society, the head of household is usually older in the family. Because only husbands and wives who were the heads of their households and their spouses were included in this study, the proportion of individuals aged 20–39 in this study is relatively low.

## 5. Conclusions

In summary, this study found that there was moderate but significant consistency in the body weight status between spouses. Although marriage generally promotes one’s health, it can also lead to poorer health and obesity, especially in vulnerable individuals, such as rural, less educated, and unstable working couples. Families with these characteristics have stronger endogenous motivation, and the family members develop less healthy behaviors and body weights. Therefore, we should identify families or couples with low social and economic levels and increase financial subsidies for the poor. Relevant departments should implement health education activities that encourage people to change behaviors and develop healthy diets and behaviors to reduce the rate of overweight or obesity, prevent obesity-related diseases, and create a healthy family living environment for the next generation.

## 6. Summary Box

### 6.1. What Is Already Known on This Topic? 

Although there is clear evidence on the presence of spousal resemblance in weight status, little is known about how family contextual factors influence the degree of spousal resemblance in BMI.

### 6.2. What Is Added by This Report? 

We analyzed the absolute and relative indicators of BMI and overweight/obesity consistency of couples to identify the characteristics of couples with high consistency. We further explored the influence of different characteristics on the level of overweight/obesity of husbands or wives and their spouses.

### 6.3. What Are the Implications for Public Health Practice?

Identification of the family contextual factors associated with spousal resemblance may help identify couples with high consistency and reduce the rate of overweight/obesity.

## Figures and Tables

**Table 1 healthcare-08-00415-t001:** Description and comparison of the basic information between the husbands and wives.

Variable		Husbands (%)	Wives (%)	χ^2^/Z	*p*-Value
BMI groups	underweight	289 (3.89)	394 (5.3)	−5.028	<0.001
	normal weight	4501 (60.56)	4663 (62.74)	
	overweight	2336 (31.43)	2044 (27.5)		
	obesity	306 (4.12)	331 (4.45)		
Overweight/Obese (BMI ≥ 25)	no	4790 (64.45)	5057 (68.04)	21.449	<0.001
	yes	2642 (35.55)	2375 (31.96)		
Obese (BMI ≥ 30)	no	7126 (95.88)	7101 (95.55)	1.025	0.311
	yes	306 (4.12)	331 (4.45)		
Age groups	20–	108 (1.45)	161 (2.17)	−5.095	<0.001
	30–	569 (7.66)	657 (8.84)		
	40–	1441 (19.39)	1460 (19.64)		
	50–	2187 (29.43)	2249 (30.26)		
	60–	2034 (27.37)	2052 (27.61)		
	70–	1093 (14.71)	853 (11.48)		
Residence	rural	5231 (70.38)	5355 (72.05)	5.047	0.025
	urban	2201 (29.62)	2077 (27.95)		
Education	illiteracy	438 (5.89)	1576 (21.21)	−23.783	<0.001
	primary school	1759 (23.67)	1940 (26.1)		
	junior middle school	3225 (43.39)	2458 (33.07)		
	senior middle school/polytechnic school/technical school	1450 (19.51)	1041 (14.01)		
	college and above	560 (7.53)	417 (5.61)		
Employment	no	3343 (44.98)	4743 (63.82)	531.565	<0.001
	yes	4089 (55.02)	2689 (36.18)		
Have children	no	826 (11.11)	930 (12.51)	6.985	0.004
	yes	6606 (88.89)	6502 (87.49)		
Current smoking	no	4017 (54.05)	7412 (99.73)	4363.955	<0.001
	yes	3415 (45.95)	20 (0.27)		
Drinking	no	4123 (55.48)	7255 (97.62)	3676.088	<0.001
	yes	3309 (44.52)	177 (2.38)		
HPA (more than once a week)	no	3570 (48.04)	3474 (46.74)	2.487	0.115
	yes	3862 (51.96)	3958 (53.26)		
Hypertension	no	5566 (74.89)	5592 (75.24)	0.243	0.622
	yes	1866 (25.11)	1840 (24.76)		
Diabetes	no	6892 (92.73)	6787 (91.32)	10.11	0.001
	yes	540 (7.27)	645 (8.68)		
Total		7432 (100)	7432 (100)		

**Table 2 healthcare-08-00415-t002:** Absolute similarity and relative association in the weight status among Chinese couples, stratified by the couple-level characteristics.

Variables	*N*	Absolute Similarity			Relative Association			
		Husbands’ BMI	Wives’ BMI	*p*	Pearson Correlation Coefficient		Kappa Coefficients	
					Partial r	95% CI	k	95% CI
Sum	7432	23.96 ± 3.33	23.72 ± 3.44	<0.001	0.102	0.076–0.120	0.049	0.031–0.069
Couples’ mean age								
20–(ref)	112	24.25 ± 3.86	22.88 ± 3.44	0.006	0.014	−0.219–0.252	0.038	−0.092–0.164
30–	564	24.82 ± 3.37	22.91 ± 3.31	<0.001	0.076	−0.008–0.167	0.053	−0.005–0.115
40–	1389	24.66 ± 3.16	23.9 ± 3.19	<0.001	0.078	0.025–0.132	0.053	0.009–0.095
50–	2116	24.03 ± 3.3	24.18 ± 3.4	0.142	0.074	0.030–0.114	0.037	0.004–0.071
60–	1958	23.53 ± 3.24	23.57 ± 3.44	0.679	0.128	0.079–0.164	0.061	0.026–0.098
70–	927	23.07 ± 3.36	23.27 ± 3.83	0.244	0.115	0.045–0.161	0.072	0.022–0.122
Residence								
both urban	1970	24.5 ± 3.25	24.01 ± 3.4	<0.001	0.078	0.033–0.12	0.036	0.002–0.072
both rural (ref)	5124	23.74 ± 3.34	23.61 ± 3.44	0.051	0.111	0.081–0.133	0.056	0.034–0.078
only husband urban	231	24.11 ± 3.12	23.83 ± 3.26	0.36	0.031	−0.102–0.163	−0.005	−0.112–0.116
only wife urban	107	24.48 ± 3.54	23.64 ± 3.94	0.1	0.118	−0.087–0.313	−0.004	−0.132–0.138
Education level								
both above primary school	3502	24.47 ± 3.18	23.71 ± 3.25	<0.001	0.05	0.017–0.083	0.04	0.016–0.068
both primary school and below (ref)	1783	23.33 ± 3.51	23.44 ± 3.63	0.374	0.15	0.101–0.19	0.075	0.037–0.111
only husband above primary school	1733	23.58 ± 3.28	24.03 ± 3.58	<0.001	0.114	0.063–0.152	0.054	0.016–0.094
only wife above primary school	414	23.96 ± 3.2	23.8 ± 3.41	0.491	0.138	0.037–0.22	0.049	−0.039–0.129
Employment status								
both employed	2391	24.04 ± 3.25	23.53 ± 3.35	<0.001	0.1	0.058–0.135	0.059	0.026–0.091
both unemployed (ref)	3045	23.77 ± 3.4	23.74 ± 3.52	0.743	0.121	0.083–0.152	0.061	0.033–0.091
only husband employed	1698	24.12 ± 3.21	24.03 ± 3.42	0.465	0.073	0.024–0.114	0.025	−0.013–0.066
only wife employed	298	24.42 ± 3.66	23.29 ± 3.3	<0.001	0.024	−0.104–0.157	-0.006	−0.091–0.082
Birth status								
only husband have children	6606	23.42 ± 3.14	23.11 ± 3.14	0.822	0.032	−0.101–0.162	0.006	−0.081–0.092
only wife have children	6502	24.58 ± 3.11	23.85 ± 3.22	0.541	0.053	0.021–0.124	0.025	−0.013–0.066
both have children	6306	24.62 ± 3.01	23.91 ± 3.41	0.003	0.022	−0.101–0.142	−0.005	−0.112–0.116
both have no children	626	23.20 ± 3.62	22.89 ± 3.01	<0.001	0.123	0.062–0.223	0.052	0.036–0.092
Current Smoking								
both smoking	11	23.12 ± 2.3	24.15 ± 5.68	0.582	−0.27	−0.417–0.191	0.035	−0.364–0.571
both nonsmoking (ref)	4008	24.14 ± 3.35	23.71 ± 3.52	<0.001	0.091	0.057–0.116	0.047	0.022–0.072
only husband smoking	3404	23.75 ± 3.28	23.73 ± 3.33	0.818	0.118	0.083–0.149	0.051	0.025–0.078
only wife smoking	9	24.49 ± 3.82	25.25 ± 3.45	0.667	0.055	−0.959–1.085	0.217	−0.333–0.751
drinking status								
both drinking	135	24.65 ± 3.18	23.86 ± 3.31	0.049	0.07	−0.112–0.251	−0.055	−0.187–0.073
both nondrinking (ref)	4081	23.62 ± 3.37	23.56 ± 3.44	0.408	0.109	0.077–0.138	0.056	0.032–0.080
only husband drinking	3174	24.37 ± 3.24	23.9 ± 3.42	<0.001	0.096	0.058–0.123	0.041	0.011–0.068
only wife drinking	42	23.88 ± 2.6	25.57 ± 3.66	0.017	−0.053	−0.332–0.257	−0.03	−0.256–0.191
Physical activity								
both HPA	2923	24.24 ± 3.21	23.71 ± 3.21	<0.001	0.09	0.054–0.126	0.063	0.033–0.093
both non-HPA (ref)	2535	23.66 ± 3.4	23.63 ± 3.59	0.825	0.119	0.075–0.148	0.044	0.011–0.075
only husband HPA	939	24.04 ± 3.36	23.84 ± 3.77	0.222	0.083	0.017–0.134	0.053	0.001–0.101
only wife HPA	1035	23.84 ± 3.37	23.85 ± 3.34	0.924	0.107	0.046–0.17	0.023	−0.025–0.072
Hypertension								
both hypertensive	660	24.55 ± 3.51	24.53 ± 3.8	0.949	0.083	0.005–0.15	0.017	−0.044–0.073
both nonhypertensive (ref)	4386	23.74 ± 3.23	23.42 ± 3.29	<0.001	0.093	0.062–0.12	0.054	0.029–0.078
only husband hypertensive	1206	25.2 ± 3.42	23.37 ± 3.38	<0.001	0.12	0.065–0.181	0.066	0.026–0.108
only wife hypertensive	1180	23.21 ± 3.1	24.73 ± 3.56	<0.001	0.119	0.054–0.154	0.043	−0.001–0.084
Diabetes								
both diabetes	82	25.57 ± 3.73	25.43 ± 4.51	0.829	0.323	0.077–0.481	0.193	0.029–0.360
both nondiabetic (ref)	6329	23.91 ± 3.33	23.63 ± 3.39	<0.001	0.098	0.072–0.12	0.046	0.025–0.066
only husband diabetic	458	24.68 ± 3.3	23.98 ± 3.47	0.002	0.135	0.041–0.222	0.015	−0.057–0.084
only wife diabetic	563	23.67 ± 3.07	24.34 ± 3.59	0.001	0.081	−0.002–0.138	0.065	−0.004–0.133

**Table 3 healthcare-08-00415-t003:** Association between overweight/obesity and an individual’s/his or her spouse’s characteristics in China.

Explanatory Variables	Husbands’ Overweight/Obesity		Wives’ Overweight/Obesity	
	OR	95% CI	OR	95% CI
The spouse’s overweight/obesity status (versus BMI < 25 kg/m^2^)	1.411	1.309–1.521	1.404	1.302–1.514
The individual’s age (per 1 years)	0.963	0.952–0.974	0.998	0.987–1.009
The spouse’s age (per 1 years)	1.003	0.992–1.015	0.993	0.982–1.004
The individual’s rural/urban residence (versus rural)	1.243	1.053–1.468	0.943	0.795–1.119
The spouse’s rural/urban residence (versus rural)	1.025	0.886–1.214	1.280	1.082–1.514
The individual’s education level (versus primary school and below)	1.032	0.944–1.128	0.742	0.680–0.809
The spouse’s education level (versus primary school and below)	1.201	1.101–1.310	1.247	1.142–1.361
The individual’s employment status (versus unemployed)	0.852	0.777–0.933	0.821	0.753–0.895
The spouse’s employment status (versus unemployed)	1.037	0.953–1.130	1.161	1.059–1.273
The individual’s childbearing situation (versus no child)	1.011	0.863–1.029	1.148	1.036–1.262
The spouse’s childbearing situation (versus no child)	1.113	1.085–1.389	1.231	1.126–1.486
The individual’s current smoking versus nonsmoking	0.684	0.634–0.737	1.226	0.626–2.400
The spouse’s current smoking versus nonsmoking	1.035	0.532–2.014	1.010	0.938–1.089
The individual’s drinking versus nondrinking	1.432	1.329–1.543	1.109	0.885–1.389
The spouse’s drinking versus nondrinking	1.126	0.900–1.408	1.205	1.118–1.298
The individual’s HPA versus (≥1 a week versus <1 a week)	1.086	1.001–1.178	0.979	0.903–1.062
The spouse’s HPA versus (≥1 a week versus <1 a week)	1.029	0.948–1.116	0.934	0.861–1.014
The individual’s hypertension versus nonhypertension	2.376	2.185–2.583	2.055	1.887–2.239
The spouse’s hypertension versus nonhypertension	1.034	0.945–1.131	0.994	0.912–1.082
The individual’s diabetes versus nondiabetes	1.198	1.047–1.372	1.098	0.969–1.244
The spouse’s diabetes versus nondiabetes	1.109	0.974–1.263	1.390	1.215–1.591

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
