# Peer review of "Socioeconomic and Demographic Factors for Spousal Resemblance in Obesity Status in China"

_healthcare, 2020, doi:10.3390/healthcare8040415_

Round 1

Reviewer 1 Report

In this interesting manuscript, Chen et al carried out an interesting study exploring the resemblance in the weight status within couples. The study is definitely extensive and logically designed, however, I am not sure about the implication of the findings. To me, the outcome of the study is sort of expected. But again, I appreciate the thorough work from the authors. In addition, I think there could be some additional parameters that, if added, could make this study more informative (please see below). If the other reviews are favorable, I would like to see the authors at least discuss about this in the discussion. This would make this study a suitable addition to Healthcare.

1. I would like to know what is the effect of having kids in the family structure? Does this make any difference? In other words, can they divide the couples in terms of having kids versus not and see the effect on the outcome. Briefly, does the size of family matters?

2. Are all the couples participated in this study are working couple? Can their job-status be a controlling factor in this study? Stress plays a huge role in obesity. Hence, could the job-related stress be a factor controlling the obesity status in these couples?

3. Finally, how do the authors take out the effects of the other parameters, such as food habit, drinking habit, exercise-status that these couples are used to in their everyday life?

Author Response

Dear Reviewing Professor:

I should like to express my appreciation to you for suggesting how to improve our paper (Title:Socioeconomic and Demographic Factors for Spousal Resemblance in Obesity Status in China.Manuscript ID:healthcare-952298).

We have carefully responded to the questions one by one according to the reviewer's requirements, and made careful revisions to the article. All revisions of the article are highlighted in red.

Question 1: I would like to know what is the effect of having kids in the family structure? Does this make any difference? In other words, can they divide the couples in terms of having kids versus not and see the effect on the outcome. Briefly, does the size of family matters?

Answer: Following your suggestion, I added the variable whether the couple has children or not in the article. After data analysis, it is found that the BMI of couples with children is greater than that of couples without children, and in couples with children, the obesity rate of the wife is higher than that of his husband. This result has also been confirmed in the relevant literature. Please review the detailed supplementary content in the article.

Question 2: Are all the couples participated in this study are working couple? Can their job-status be a controlling factor in this study? Stress plays a huge role in obesity. Hence, could the job-related stress be a factor controlling the obesity status in these couples?

Answer: Not all couples in this survey are working couple,Table 2 divides the participants' work status into whether they are employed or not. The results showed that 55.02% of husbands were employed and 36.18% of wives were employed. The results in Table 3 show that:Whether it’s a wife or a husband, those who are employed have lower rates of overweight or obesity than those who are unemployed. Wives whose spouses had jobs were 1.161 times more likely to be overweight or obese than those whose spouses were unemployed. Relevant research and literature show that stress does cause overweight or obesity. This is described in lines 220-221 of the article.

Question 3: Finally, how do the authors take out the effects of the other parameters, such as food habit, drinking habit, exercise-status that these couples are used to in their everyday life?

Answer: The habit parameters of couples in daily life, such as smoking status, drinking habits, and exercise status are all dichotomy issues, reported by the household respondent to the following questions: "Do you smoke now?" "Have you drink alcohol or have had alcoholic drinks in the past 12 months?" "Do you now spend half an hour or more in moderate or vigorous physical activity at least three times a week?"This question has been revised and supplemented in lines 84-88 of the article.

Because of your suggestions, the revised article has become better, and readers can obtain more valuable information. Thanks again to the editors and reviewers for their help.If you have any question about this paper,please don’t hesitate to let me know.

Best wishes,

Sincerely yours,

Xuejiao Chen

Reviewer 2 Report

Overall this is a well written article. I have made the following observations as I have read through it;

Line 25: Check sentence structure – not clearly expressed.

Line 148: Unsure about the value of Table 2. Too much information Would be better to have a summary table showing key findings.

Line 163/164: “For a wife, the more educated she was, the less likely she was to be overweight or obese, 163 while the more educated a spouse was, the more likely he or she was to be overweight or obese.” This appears to be contradictory – a wife is a spouse but if the second part of the statement is referring to her spouse, then it cannot be ‘he or she’. Need a clearer explanation.

Line 171: Review “In couples involves a husband who drank alcohol”

Line 173: Unsure why ‘only’ is in this sentence. One would expect those who exercised would be less overweight. “Husbands who exercised more than once a week were only 8.6% 172 more likely to be overweight or obese than those who did not exercise more than once a week.”

Line 210: Requires a reference. “Studies have shown that men who smoke have a lower risk of becoming overweight or obese than men who do not smoke, but drinking alcohol increases the risk of being overweight or obese for married men themselves and their spouses.”

Line 212: There is no mention that central obesity was measured as part of this study. If this is a general statement, then it needs to be referenced. “In general, married men who have drinking habits tend to become overweight or obese or even have central obesity.”

Line 213: Requires a reference. “Because of social interactions or pressure, most of these men have bad eating habits, such as excessive oil and salt intake, overeating or an irregular diet.”

Line 215: Requires referencing. “According to previous studies, people who are overweight or obese are considered to be at high risk of hypertension, hyperglycemia and hyperlipidemia, which are usually accompanied by obesity.”

My main concern is with the conclusion. The authors seems to ignore the fact that poverty and food insecurity are strongly associated with obesity, and assumes that these individuals are overweight because of a lack of education. One of the weaknesses that the authors identify is the they were unable to look at causality. It is therefore inappropriate to assume that education is the key to addressing the issues of obesity.

Author Response

Dear Reviewing Professor:

I should like to express my appreciation to you for suggesting how to improve our paper (Title:Socioeconomic and Demographic Factors for Spousal Resemblance in Obesity Status in China.Manuscript ID:healthcare-952298).

We have carefully responded to the questions one by one according to the reviewer's requirements, and made careful revisions to the article. All revisions of the article are highlighted in red.

Question 1: Line 25: Check sentence structure – not clearly expressed.

Answer: Modify the sentence in the manuscript as follows:Health education activities to couples for intervention, can have a good effect of intervention.

Question 2:Line 148: Unsure about the value of Table 2. Too much information Would be better to have a summary table showing key findings.

Answer: Thank you very much for your question. You proposed to add a summary table showing key findings. After checking the table, I found that most of the variables in the table are meaningful, and only a few variables can be deleted. There are indeed many variables in Table 2, so I bolded the results meaningfully for everyone to read, and briefly described the key information in the results section. Do you think this is fine? If you do not agree, I am willing to modify it again to meet your satisfaction.

Question3:Line 163/164: “For a wife, the more educated she was, the less likely she was to be overweight or obese, 163 while the more educated a spouse was, the more likely he or she was to be overweight or obese.” This appears to be contradictory – a wife is a spouse but if the second part of the statement is referring to her spouse, then it cannot be ‘he or she’. Need a clearer explanation.

Answer: I am very sorry for the inconsistency of the sentence due to my unclear expression. This sentence should have two parts. The first part is for all wives. The higher education level among all wives, the less overweight or obese; the second part is for husband and wife, the higher the education of the wife or husband , His spouse is more likely to be obese or overweight.Therefore, I modify it as follows:Among all wives in this survey, the more educated she was, the less likely she was to be overweight or obese. Among couples, the higher the education level of the husband or wife,  the more likely their spouse was to be overweight or obese.

Question 4:Line 171: Review “In couples involves a husband who drank alcohol”

Answer: In accordance with your request, I have supplemented and modified this sentence as follows:In couples involves a husband who drank alcohol, the wives risk of being overweight or obese increases by 20.5%.

Question 5:Line 173: Unsure why ‘only’ is in this sentence. One would expect those who exercised would be less overweight. “Husbands who exercised more than once a week were only 8.6% 172 more likely to be overweight or obese than those who did not exercise more than once a week.”

Answer: I think your suggestion is correct, so the word "only" is deleted.

Question 6:Line 210: Requires a reference. “Studies have shown that men who smoke have a lower risk of becoming overweight or obese than men who do not smoke, but drinking alcohol increases the risk of being overweight or obese for married men themselves and their spouses.”

Answer: According to your request, add reference 27 at the end of this sentence.

Question 7:Line 212: There is no mention that central obesity was measured as part of this study. If this is a general statement, then it needs to be referenced. “In general, married men who have drinking habits tend to become overweight or obese or even have central obesity.”

Answer: I think central obesity is unnecessary in this article, so I delete the sentence "or even have central obesity".

Question 8:Line 213: Requires a reference. “Because of social interactions or pressure, most of these men have bad eating habits, such as excessive oil and salt intake, overeating or an irregular diet.”

Answer: Add reference 28 at the end of this sentence.

Question 9:Line 215: Requires referencing. “According to previous studies, people who are overweight or obese are considered to be at high risk of hypertension, hyperglycemia and hyperlipidemia, which are usually accompanied by obesity.”

Answer: Add reference 29 at the end of this sentence.

Question 10:My main concern is with the conclusion. The authors seems to ignore the fact that poverty and food insecurity are strongly associated with obesity, and assumes that these individuals are overweight because of a lack of education. One of the weaknesses that the authors identify is the they were unable to look at causality. It is therefore inappropriate to assume that education is the key to addressing the issues of obesity.

Answer: I am very sorry for ignoring the two issues of poverty and food security. After consulting the relevant literature, I found that the two issues you raised are closely related to overweight and obesity. Therefore, I added a description of poverty and food security in the conclusion section, suggesting that relevant departments not only prevent obesity in education, but also increase financial subsidies to the poor and pay attention to food security.

Because of your suggestions, the revised article has become better, and readers can obtain more valuable information. Thanks again to the editors and reviewers for their help.If you have any question about this paper,please don’t hesitate to let me know.

Best wishes,

Sincerely yours,

Xuejiao Chen
